# When Does Label Smoothing Help?

**Rafael Müller,**\* **Simon Kornblith, Geoffrey Hinton**
Google Brain
Toronto
`rafaelmuller@google.com`

## Abstract

The generalization and learning speed of a multi-class neural network can often be significantly improved by using soft targets that are a weighted average of the hard targets and the uniform distribution over labels. Smoothing the labels in this way prevents the network from becoming over-confident and label smoothing has been used in many state-of-the-art models, including image classification, language translation and speech recognition. Despite its widespread use, label smoothing is still poorly understood. Here we show empirically that in addition to improving generalization, label smoothing improves model calibration which can significantly improve beam-search. However, we also observe that if a teacher network is trained with label smoothing, knowledge distillation into a student network is much less effective. To explain these observations, we visualize how label smoothing changes the representations learned by the penultimate layer of the network. We show that label smoothing encourages the representations of training examples from the same class to group in tight clusters. This results in loss of information in the logits about resemblances between instances of different classes, which is necessary for distillation, but does not hurt generalization or calibration of the model's predictions.

## 1   Introduction

It is widely known that neural network training is sensitive to the loss that is minimized. Shortly after Rumelhart et al. [1] derived backpropagation for the quadratic loss function, several researchers noted that better classification performance and faster convergence could be attained by performing gradient descent to minimize cross entropy [2, 3]. However, even in these early days of neural network research, there were indications that other, more exotic objectives could outperform the standard cross entropy loss [4, 5]. More recently, Szegedy et al. [6] introduced label smoothing, which improves accuracy by computing cross entropy not with the "hard" targets from the dataset, but with a weighted mixture of these targets with the uniform distribution.

Label smoothing has been used successfully to improve the accuracy of deep learning models across a range of tasks, including image classification, speech recognition, and machine translation (Table 1). Szegedy et al. [6] originally proposed label smoothing as a strategy that improved the performance of the Inception architecture on the ImageNet dataset, and many state-of-the-art image classification models have incorporated label smoothing into training procedures ever since [7, 8, 9]. In speech recognition, Chorowski and Jaitly [10] used label smoothing to reduce the word error rate on the WSJ dataset. In machine translation, Vaswani et al. [11] attained a small but important improvement in BLEU score, despite a reduction in perplexity.

Although label smoothing is a widely used "trick" to improve network performance, not much is known about why and when label smoothing should work. This paper tries to shed light upon behavior

Table 1: Survey of literature label smoothing results on three supervised learning tasks.

| DATA SET | ARCHITECTURE | METRIC | VALUE W/O LS | VALUE W/ LS |
|---|---|---|---|---|
| IMAGENET | INCEPTION-V2 [6] | TOP-1 ERROR | 23.1 | **22.8** |
| | | TOP-5 ERROR | 6.3 | **6.1** |
| EN-DE | TRANSFORMER [11] | BLEU | 25.3 | **25.8** |
| | | PERPLEXITY | **4.67** | 4.92 |
| WSJ | BILSTM+ATT.[10] | WER | 8.9 | 7.0/**6.7** |

of neural networks trained with label smoothing, and we describe several intriguing properties of these networks. Our contributions are as follows:

- We introduce a novel visualization method based on linear projections of the penultimate layer activations. This visualization provides intuition regarding how representations differ between penultimate layers of networks trained with and without label smoothing.

- We demonstrate that label smoothing implicitly calibrates learned models so that the confidences of their predictions are more aligned with the accuracies of their predictions.

- We show that label smoothing impairs distillation, *i.e.*, when teacher models are trained with label smoothing, student models perform worse. We further show that this adverse effect results from loss of information in the logits.

## 1.1 Preliminaries

Before describing our findings, we provide a mathematical description of label smoothing. Suppose we write the prediction of a neural network as a function of the activations in the penultimate layer as $p_k = \frac{e^{\boldsymbol{x}^T \boldsymbol{w}_k}}{\sum_{l=1}^{L} e^{\boldsymbol{x}^T \boldsymbol{w}_l}}$, where $p_k$ is the likelihood the model assigns to the $k$-th class, $\boldsymbol{w}_k$ represents the weights and biases of the last layer, $\boldsymbol{x}$ is the vector containing the activations of the penultimate layer of a neural network concatenated with "1" to account for the bias. For a network trained with hard targets, we minimize the expected value of the cross-entropy between the true targets $y_k$ and the network's outputs $p_k$ as in $H(\boldsymbol{y}, \boldsymbol{p}) = \sum_{k=1}^{K} -y_k \log(p_k)$, where $y_k$ is "1" for the correct class and "0" for the rest. For a network trained with a label smoothing of parameter $\alpha$, we minimize instead the cross-entropy between the modified targets $y_k^{LS}$ and the networks' outputs $p_k$, where $y_k^{LS} = y_k(1 - \alpha) + \alpha/K$.

## 2 Penultimate layer representations

Training a network with label smoothing encourages the differences between the logit of the correct class and the logits of the incorrect classes to be a constant dependent on $\alpha$. By contrast, training a network with hard targets typically results in the correct logit being much larger than the any of the incorrect logits and also allows the incorrect logits to be very different from one another. Intuitively, the logit $\boldsymbol{x}^T \boldsymbol{w}_k$ of the k-th class can be thought of as a measure of the squared Euclidean distance between the activations of the penultimate layer $\boldsymbol{x}$ and a template $\boldsymbol{w}_k$, as $||\boldsymbol{x} - \boldsymbol{w}_k||^2 = \boldsymbol{x}^T \boldsymbol{x} - 2\boldsymbol{x}^T \boldsymbol{w}_k + \boldsymbol{w}_k^T \boldsymbol{w}_k$. Here, each class has a template $\boldsymbol{w}_k$, $\boldsymbol{x}^T \boldsymbol{x}$ is factored out when calculating the softmax outputs and $\boldsymbol{w}_k^T \boldsymbol{w}_k$ is usually constant across classes. Therefore, *label smoothing encourages the activations of the penultimate layer to be close to the template of the correct class and equally distant to the templates of the incorrect classes*. To observe this property of label smoothing, we propose a new visualization scheme based on the following steps: (1) Pick three classes, (2) Find an orthonormal basis of the plane crossing the templates of these three classes, (3) Project the penultimate layer activations of examples from these three classes onto this plane. This visualization shows in 2-D how the activations cluster around the templates and how label smoothing enforces a structure on the distance between the examples and the clusters from the other classes.

In Fig. 1, we show results of visualizing penultimate layer representations of image classifiers trained on the datasets CIFAR-10, CIFAR-100 and ImageNet with the architectures AlexNet [12], ResNet-56 [13] and Inception-v4 [14], respectively. Table 2 shows the effect of label smoothing on the accuracy of these models. We start by describing visualization results for CIFAR-10 (first row of Fig. 1) for the classes "airplane," "automobile" and "bird." The first two columns represent examples from the training and validation set for a network trained without label smoothing (w/o LS). We observe that

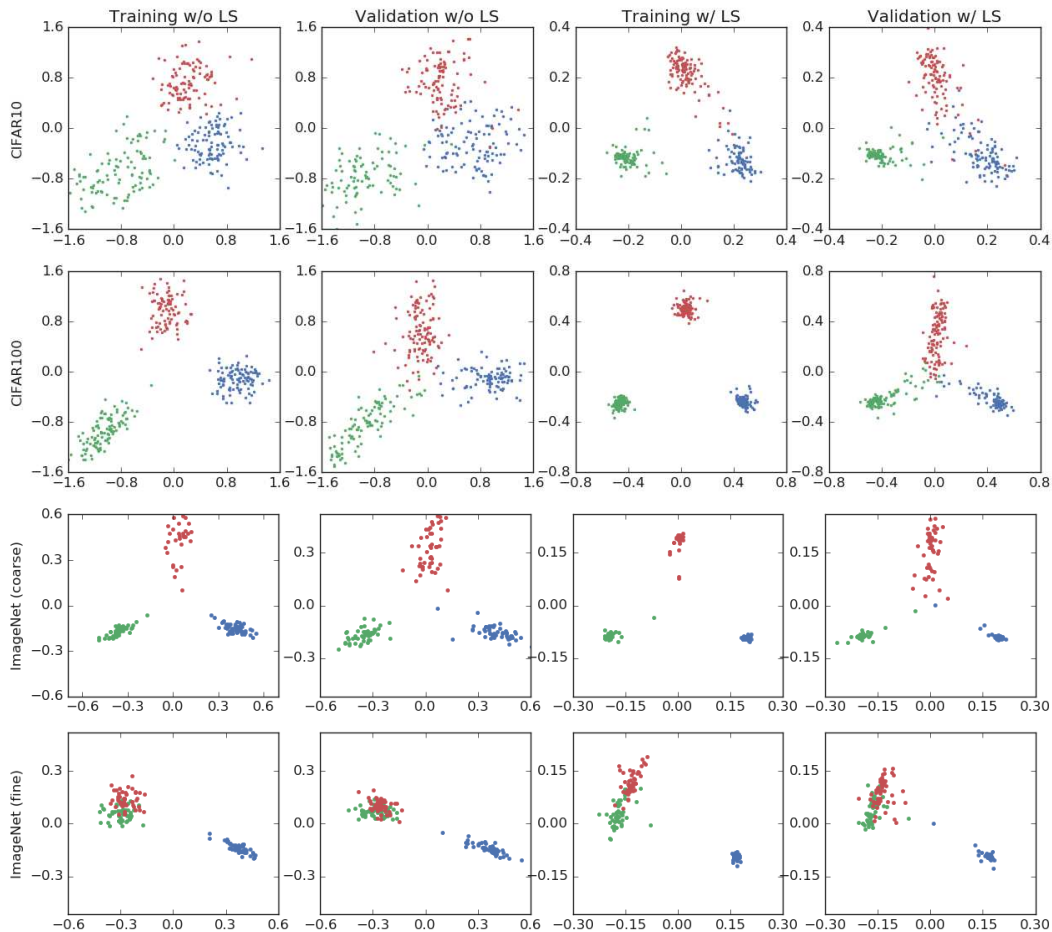

Figure 1: Visualization of penultimate layer's activations of: AlexNet/CIFAR-10 (first row), CIFAR-100/ResNet-56 (second row) and ImageNet/Inception-v4 with three semantically different classes (third row) and two semantically similar classes plus a third one (fourth row).

Table 2: Top-1 classification accuracies of networks trained with and without label smoothing used in visualizations.

| DATA SET | ARCHITECTURE | ACCURACY ($\alpha = 0.0$) | ACCURACY ($\alpha = 0.1$) |
|---|---|---|---|
| CIFAR-10 | ALEXNET | $86.8 \pm 0.2$ | $86.7 \pm 0.3$ |
| CIFAR-100 | RESNET-56 | $72.1 \pm 0.3$ | $72.7 \pm 0.3$ |
| IMAGENET | INCEPTION-V4 | $80.9$ | $80.9$ |

the projections are spread into defined but broad clusters. The last two columns show a network trained with a label smoothing factor of 0.1. We observe that now the clusters are much tighter, because label smoothing encourages that each example in training set to be equidistant from all the other class's templates. Therefore, when looking at the projections, the clusters organize in regular triangles when training with label smoothing, whereas the regular triangle structure is less discernible in the case of training with hard-targets (no label smoothing). Note that these networks have similar accuracies despite qualitatively different clustering of activations.

In the second row, we investigate the activation's geometry for a different pair of dataset/architecture (CIFAR-100/ResNet-56). Again, we observe the same behavior for classes "beaver," "dolphin," "otter." In contrast to the previous example, now the networks trained with label smoothing have better accuracy. Additionally, we observe the different scale of the projections between the network trained with and without label smoothing. With label smoothing, the difference between logits of two classes has to be limited in absolute value to get the desired soft target for the correct and incorrect

classes. Without label smoothing, however, the projection can take much higher absolute values, which represent over-confident predictions.

Finally, we test our visualization scheme in an Inception-v4/ImageNet experiment and observe the effect of label smoothing for semantically similar classes, since ImageNet has many fine-grained classes (e.g. different breeds of dogs). The third row represents projections for semantically different classes (tench, meerkat and cleaver) with the behavior similar to previous experiments. The fourth row is more interesting, since we pick two semantically similar classes (toy poodle and miniature poodle) and observe the projection with the presence of a third semantically different one (tench, in blue). With hard targets, the semantically similar classes cluster close to each other with an isotropic spread. On the contrary, with label smoothing these similar classes lie in an arc. In both cases, the semantically similar classes are harder to separate even on the training set, but label smoothing enforces that each example be equidistant to all remaining class's templates, which gives rise to arc shape behavior with respect to other classes. We also observe that when training without label smoothing there is continuous degree of change between the "tench" cluster and the "poodles" cluster. We can potentially measure "how much a poodle is a particular tench". However, when training with label smoothing this information is virtually erased. This erasure of information is shown in the Section 4. Finally, the figure shows that the effect of label smoothing on representations is independent of architecture, dataset and accuracy.

## 3 Implicit model calibration

By artificially softening the targets, label smoothing prevents the network from becoming over-confident. But does it improve the calibration of the model by making the confidence of its predictions more accurately represent their accuracy? In this section, we seek to answer this question. Guo et al. [15] have shown that modern neural networks are poorly calibrated and over-confident despite having better performance than better calibrated networks from the past. To measure calibration, the authors computed the estimated expected calibration error (ECE). They demonstrated that a simple post-processing step, temperature scaling, can reduce ECE and calibrate the network. Temperature scaling consists in multiplying the logits by a scalar before applying the softmax operator. Here, we show that label smoothing also reduces ECE and can be used to calibrate a network without the need for temperature scaling.

**Image classification.** We start by investigating the calibration of image classification models. Fig. 2 (left) shows the 15-bin reliability diagram of a ResNet-56 trained on CIFAR-100. The dashed line represent perfect calibration where the output likelihood (confidence) predicts perfectly the accuracy. Without temperature scaling, the model trained with hard targets (blue line without markers) is clearly over-confident, since in expectation the accuracy is always below the confidence. To calibrate the model, one can tune the softmax temperature a posteriori (blue line with crosses) to a temperature of 1.9. We observe that the reliability diagram slope is now much closer to a slope of 1 and the model is better calibrated. We also show that, in terms of calibration, label smoothing has a similar effect. By training the same model with $\alpha = 0.05$ (green line), we obtain a model that is similarly calibrated compared to temperature scaling. In Table 3, we observe how varying label smoothing and temperature scaling affects ECE. Both methods can be used to reduce ECE to a similar and smaller value than an uncalibrated network trained with hard targets.

We also performed experiments on ImageNet (Fig. 2 right). Again, the network trained with hard targets (blue curve without markers) is over-confident and achieves a high ECE of 0.071. Using temperature scaling (T= 1.4), ECE is reduced to 0.022 (blue curve with crosses). Although we did not tune $\alpha$ extensively, we found that label smoothing of $\alpha = 0.1$ improves ECE to 0.035, resulting in better calibration compared to the unscaled network trained with hard targets.

These results are somewhat surprising in light of the penultimate layer visualizations of these networks shown in the previous section. Despite trying to collapse the training examples to tiny clusters, these networks generalize and are calibrated. Looking at the label smoothing representations for CIFAR-100 in Fig. 1 (second row, two last columns), we clearly observe this behavior. The red cluster is very tight in the training set, but in the validation set it spreads towards the center representing the full range of confidences for each prediction.

**Machine translation.** We also investigate the calibration of a model trained on the English-to-German translation task using the Transformer architecture. This setup is interesting for two reasons. First, Vaswani et al. [11] noted that label smoothing with $\alpha = 0.1$ improved the BLEU score of

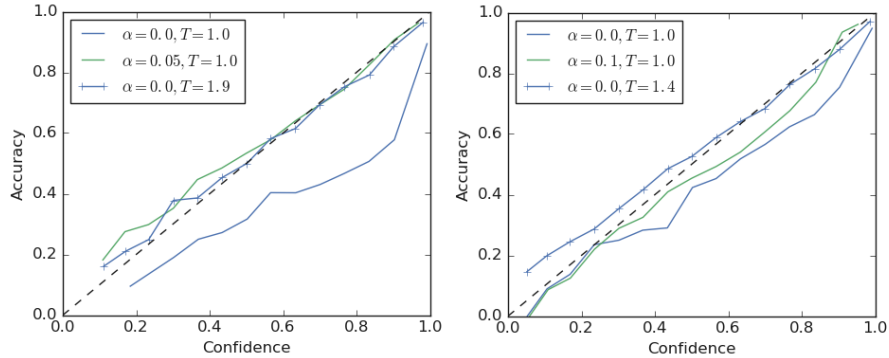

Figure 2: Reliability diagram of ResNet-56/CIFAR-100 (left) and Inception-v4/ImageNet (right).

Table 3: Expected calibration error (ECE) on different architectures/datasets.

| DATA SET | ARCHITECTURE | BASELINE ECE (T=1.0, $\alpha = 0.0$) | TEMP. SCALING ECE / T ($\alpha = 0.0$) | LABEL SMOOTHING ECE / $\alpha$ (T=1.0) |
|---|---|---|---|---|
| CIFAR-100 | RESNET-56 | 0.150 | 0.021 / 1.9 | 0.024 / 0.05 |
| IMAGENET | INCEPTION-V4 | 0.071 | 0.022 / 1.4 | 0.035 / 0.1 |
| EN-DE | TRANSFORMER | 0.056 | 0.018 / 1.13 | 0.019 / 0.1 |

their final model despite attaining worse perplexity compared to a model trained with hard targets ($\alpha = 0.0$). So we compare both setups in terms of calibration to verify that label smoothing also improves calibration in this task. Second, compared to image classification, where calibration does not directly affect the metric we care about (accuracy), in language translation, the network's soft outputs are inputs to a second algorithm (beam-search) which is affected by calibration. Since beam-search approximates a maximum likelihood sequence detection algorithm (Viterbi algorithm), we would intuitively expect better performance for a better calibrated model, since the model's confidence predicts better the accuracy of the next token.

We start by looking at the reliability diagram (Fig. 3) for a Transformer network trained with hard targets (with and without temperature scaling) and a network trained with label smoothing ($\alpha = 0.1$). We plot calibration of the next-token predictions assuming a correct prefix on the validation set. The results are in agreement with the previous experiments on CIFAR-100 and ImageNet, and indeed the Transformer network [11] with label smoothing is better calibrated than the hard targets alternative.

Despite being better calibrated and achieving better BLEU scores, label smoothing results in worse negative log-likelihoods (NLL) than hard targets. Moreover, temperature scaling with hard targets is insufficient to recover the BLEU score improvement obtained with label smoothing. In Fig. 4, we artificially vary calibra-

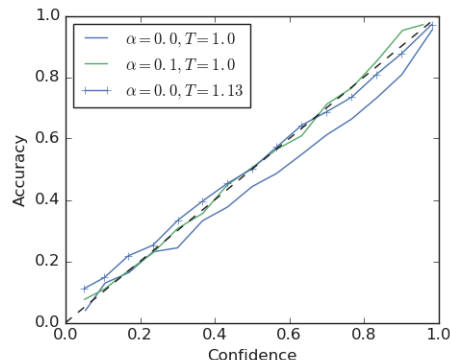

Figure 3: Reliability diagram of Transformer trained on EN-DE dataset.

tion of both networks, using temperature scaling, and analyze the effect upon BLEU and NLL. The left panel shows results for a network trained with hard targets. By increasing the temperature we can both reduce ECE (red, right y-axis) and slightly improve BLEU score (blue left y-axis), but the BLEU score improvement is not enough to match the BLEU score of the network trained with label smoothing (center panel). The network trained with label smoothing is "automatically calibrated" and changing the temperature degrades both calibration and BLEU score. Finally, in the right panel, we plot the NLL for both networks, where markers represent the network with label smoothing. The model trained with hard targets achieves better NLL at all temperature scaling settings. Thus, label smoothing improves translation quality measured by BLEU score despite worse NLL, and the difference of BLEU score performance is only partly explained by calibration. Note that the minimum of ECE for this experiment predicts top BLEU score slightly better than NLL.

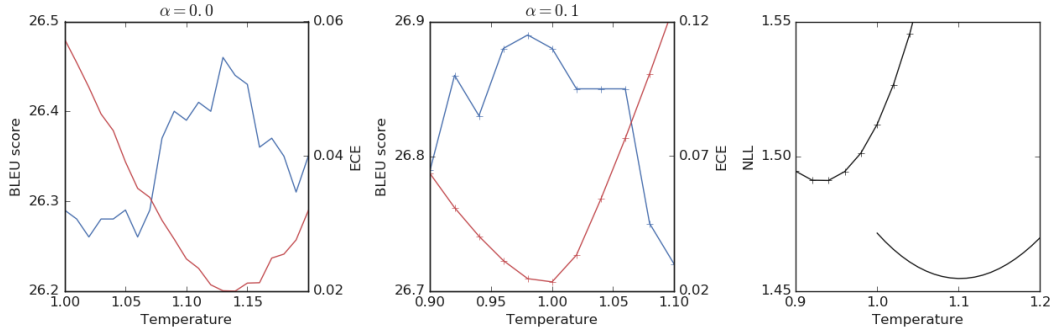

Figure 4: Effect of calibration of Transformer upon BLEU score (blue lines) and NLL (red lines). Curves without markers reflect networks trained without label smoothing while curves with markers represent networks with label smoothing.

# 4 Knowledge distillation

In this section, we study how the use of label smoothing to train a teacher network affects the ability to distill the teacher's knowledge into a student network. We show that, even when label smoothing improves the accuracy of the teacher network, teachers trained with label smoothing produce inferior student networks compared to teachers trained with hard targets. We first noticed this effect when trying to replicate a result in [16]. A non-convolutional teacher is trained on randomly translated MNIST digits with hard targets and dropout and gets 0.67% test error. Using distillation, this teacher can be used to train a narrower, unregularized student on untranslated digits to get 0.74% test error. If we use label smoothing instead of dropout, the teacher trains much faster and does slightly better (0.59%) but distillation produces a much worse student with 0.91% test error. Something goes seriously wrong with distillation when the teacher is trained with label smoothing.

In knowledge distillation, we replace the cross-entropy term $H(\boldsymbol{y}, \boldsymbol{p})$ by the weighted sum $(1 - \beta)H(\boldsymbol{y}, \boldsymbol{p}) + \beta H(\boldsymbol{p}^t(T), \boldsymbol{p}(T))$, where $p_k(T)$ and $p_k^t(T)$ are the outputs of the student and teacher after temperature scaling with temperature $T$, respectively. $\beta$ controls the balance between two tasks: fitting the hard-targets and approximating the softened teacher. The temperature can be viewed as a way to exaggerate the differences between the probabilities of incorrect answers.

Both label smoothing and knowledge distillation involve fitting a model using soft targets. Knowledge distillation is only useful if it provides an additional gain to the student compared to training the student with label smoothing, which is simpler to implement since it does not require training a teacher network. We quantify this gain experimentally. To demonstrate these ideas, we perform an experiment on the CIFAR-10 dataset. We train a ResNet-56 teacher and we distill to an AlexNet student. We are interested in four results:

1. the teacher's accuracy as a function of the label smoothing factor,
2. the student's baseline accuracy as a function of the label smoothing factor without distillation,
3. the student's accuracy after distillation with temperature scaling to control the smoothness of the teacher's provided targets (teacher trained with hard targets)
4. the student's accuracy after distillation with fixed temperature ($T = 1.0$ and teacher trained with label smoothing to control the smoothness of the teacher's provided targets)

To compare all solutions using a single smoothness index, we define the equivalent label smoothing factor $\gamma$ which for scenarios 1 and 2 are equal to $\alpha$. For scenarios 3 and 4, the smoothness index is $\gamma = \mathbb{E}\big[\sum_{k=1}^{K}(1 - y_k)p_k^t(T)K/(K - 1)\big]$, which calculates the mass allocated by the teacher to incorrect examples over the training set. Since the training accuracy is nearly perfect, for all distillation experiments, we consider only the case where $\beta = 1$, *i.e.*, when the targets are the teacher output and the true labels are ignored.

Fig. 5 shows the results of this distillation experiment. We first compare the performance of the teacher network (blue solid curve, top) and student network (light blue solid, bottom) trained without distillation. For this particular setup, increasing $\alpha$ improves the teacher's accuracy up to values of $\alpha = 0.6$, while label smoothing slightly degrades the baseline performance of the student networks.

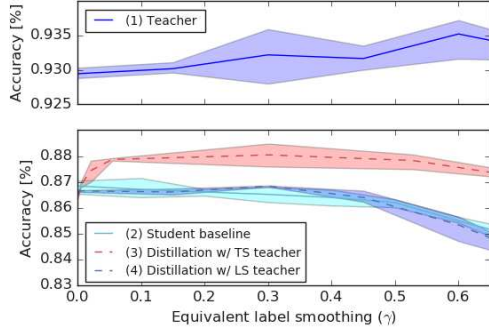

Figure 5: Performance of distillation from ResNet-56 to AlexNet on CIFAR-10.

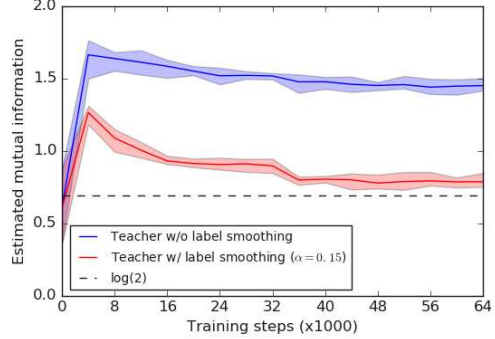

Figure 6: Estimated mutual information evolution during teacher training.

Next, we distill the teacher trained with hard targets to students using different temperatures, and calculate the corresponding $\gamma$ for each temperature (red dashed curve). We observe that all distilled models outperform the baseline student with label smoothing. Finally, we distill information from teachers trained with label smoothing $\alpha > 0$, which have better accuracy (blue dashed curve). The figure shows that using these better performing teachers is no better, and sometimes worse, than training the student directly with label smoothing, as the relative information between logits is "erased" when the teacher is trained with label smoothing.

To observe how label smoothing "erases" the information contained in the different similarities that an individual example has to different classes we revisit the visualizations in Fig. 1. Note that we are interested in the visualization of the examples from the training set, since those are the ones used for distillation. For a teacher trained with hard targets ($\alpha = 0.0$), we observe that examples are distributed in broad clusters, which means that different examples from the same class can have very different similarities to other classes. For a teacher trained with label smoothing, we observe the opposite behavior. Label smoothing encourages examples to lie in tight equally separated clusters, so every example of one class has very similar proximities to examples of the other classes. Therefore, *a teacher with better accuracy is not necessarily the one that distills better*.

One way to directly quantify information erasure is to calculate the mutual information between the input and the logits. Computing mutual information in high dimensions for unknown distributions is challenging, but here we simplify the problem in several ways. We measure the mutual information between $X$ and $Y$, where $X$ is a discrete variable representing the index of the training example and $Y$ is continuous representing the difference between two logits (out of K classes). The source of randomness comes from data augmentation and we approximate the distribution of $Y$ as a Gaussian and we estimate the mean and variance from the examples using Monte Carlo samples. The difference of the logits $y$ can be written as $y = f(d(\boldsymbol{z}_x))$, where $\boldsymbol{z}_x$ is the flattened input image indexed by $x$, $d(\cdot)$ is a random data augmentation function (random shifts for example), and $f(\cdot)$ is a trained neural network with an image as an input and the difference between two logits as output (resulting in a single dimension real-valued output). The mutual information and its respective approximation are $I(X; Y) = E_{X,Y}[\log(p(y|x)) - \log(\sum_x p(y|x))]$ and

$$\hat{I}(X; Y) = \sum_{x=1}^{N} \big[ - (f(d(\boldsymbol{z}_x)) - \mu_x)^2/(2\sigma^2) - \log(\sum_{x=1}^{N} e^{-(f(d(\boldsymbol{z}_x)) - \mu_x)^2/(2\sigma^2)}) \big],$$

where $\mu_x = \sum_{l=1}^{L} f(d(\boldsymbol{z}_x))/L$, $\sigma^2 = \sum_{x=1}^{N} (f(d(\boldsymbol{z}_x)) - \mu_x)^2/N$, $L$ is the number of Monte Carlo samples used to calculate the empirical mean and $N$ is the number of training examples used for mutual information estimation. Here the mutual information is between 0 and $\log(N)$.

Fig. 6 shows the estimated mutual information between a subset (N = 600 from two classes) of the training examples and the difference of the logits corresponding to these two classes. After initialization, the mutual information is very small, but as the network is trained, first it rapidly increases then it slowly decreases specially for the network trained with label smoothing. This result confirms the intuitions from the last sections. As the representations collapse to small clusters of points, much of the information that could have helped distinguish examples is lost. This results in lower estimated mutual information and poor distillation for teachers trained with label smoothing.

For later stages of training, mutual information stays slightly above $\log(2)$, which corresponds to the extreme case where all training examples collapse to two separate clusters. In this case, all the information of the input is discarded except a single bit representing which class the example belongs to, resulting in no extra information in the teacher's logits compared to the information in the labels.

## 5 Related work

Pereyra et al. [17] showed that label smoothing provides consistent gains across many tasks. That work also proposed a new regularizer based on penalizing low entropy predictions, which the authors term "confidence penalty." They show that label smoothing is equivalent to confidence penalty if the order of the KL divergence between uniform distributions and model's outputs is reversed. They also propose to use distributions other than uniform, resulting in unigram label smoothing (see Table 1) which is advantageous when the output labels' distribution is not balanced. Label smoothing also relates to DisturbLabel [18], which can be seen as label dropout, whereas label smoothing is the marginalized version of label dropout.

Calibration of modern neural networks [15] has been widely investigated for image classification, but calibration of sequence models has been investigated only recently. Ott et al. [19] investigate the sequence level calibration of machine translation models and conclude they are remarkably well calibrated. Kumar and Sarawagi [20] investigate calibration of next-token prediction in language translation. They find that calibration of state-of-the-art models can be improved by a parametric model, resulting in a small increase in BLEU score. However, neither work invesigates the relation between label smoothing during training and calibration. For speech recognition, Chorowski and Jaitly [10] investigate the effect of softmax temperature and label smoothing on decoding accuracy. The authors conclude that both temperature scaling and label smoothing improve word error rates after beam-search (label smoothing performs best), but the relation between calibration and label smoothing/temperature scaling is not described.

Although we are unaware of any previous work that shows the adverse effect of label smoothing upon distillation, Kornblith et al. [21] previously demonstrated that label smoothing impairs the accuracy of transfer learning, which similarly depends on the presence of non-class-relevant information in the final layers of the network. Additionally, Chelombiev et al. [22] propose an improved mutual information estimator based on binning and show correlation between compression of softmax layer representations and generalization, which may explain why networks trained with label smoothing generalize so well. This relates to the information bottleneck theory [23, 24, 25] that explains generalization in terms of compression.

## 6 Conclusion and future work

Many state-of-the-art models are trained with label smoothing, but the inductive bias provided by this technique is not well understood. In this work, we have summarized and explained several behaviors observed while training deep neural networks with label smoothing. We focused on how label smoothing encourages representations in the penultimate layer to group in tight equally distant clusters. This emergent property can be visualized in low dimension thanks to a new visualization scheme that we proposed. Despite having a positive effect on generalization and calibration, label smoothing can hurt distillation. We explain this effect in terms of erasure of information. With label smoothing, the model is encouraged to treat each incorrect class as equally probable. With hard targets, less structure is enforced in later representations, enabling more logit variation across predicted class and/or across examples. This can be quantified by estimating mutual information between input example and output logit and, as we have shown, label smoothing reduces mutual information. This finding suggests a new research direction, focusing on the relationship between label smoothing and the information bottleneck principle, with implications for compression, generalization and information transfer. Finally, we performed extensive experiments on how label smoothing can implicitly calibrate model's predictions. This has big impact on model interpretability, but also, as we have shown, can be critical for downstream tasks that depend on calibrated likelihoods such as beam-search.

## 7 Acknowledgements

We would like to thank Mohammad Norouzi, William Chan, Kevin Swersky, Danijar Hafner and Rishabh Agrawal for the discussions and suggestions.

## Footnotes

\*This work was done as part of the Google AI Residency.

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
