[Supplementary Material · appendix.pdf]

# Appendix

## A   Experimental details

### A.1   AlexNet

The AlexNet architecture was used in the visualization section and in the distillation section. In both cases, it was trained on the CIFAR-10 dataset. We split the training set into training and validation (40k/10k) and results are shown in the validation set. Our implementation is based on the public available code in [2] and [3] without exponential weight averaging. The mini-batch size is 128 and we train for 390k iterations with stochastic gradient descent, starting with learning rate 0.1 and droping by a factor of 10 every 130k steps. We multiply the cross-entropy term by 3 and use weight decay of 0.04 in the last two dense layers. Five different seeds were used for each point shown in the figure and we show min/average/max. For visualization of representations of CIFAR-10 a total of 300 examples were used (100 per class). For the distillation results, $\beta$ is set to 1 so distillation is done fully with teacher targets. Note that we use the same training procedure for: training the student without distillation (variable label smoothing), training the student to mimic the teacher trained with hard targets and temperature scaling, and training the student to mimic the teacher trained with variable label smoothing. In the end, the only difference is the targets used. Additionally, when $\beta = 1$, the temperature of the teacher is important to soften the targets, but the temperature of the student ends up to be just a scalar multiplying the logits and can be learned, therefore we set the student soft-max temperature to 1 for all experiments. For distillation with teacher trained with hard targets, the temperatures tested were 1.0, 2.0, 3.0, 4.0, 8.0, 12.0 and 16.0. The label smoothing values used were 0.0 to 0.75 with steps of 0.15.

### A.2   ResNet

The ResNet architecture was used in the visualization, calibration and distillation sections. In all cases, it was trained on the CIFAR-10 and CIFAR-100 datasets[4]. We split the training set into training and validation (40k/10k) and results are shown in the validation set. Our implementation uses the public available model in [5]. The mini-batch size is 128 and we train for 64k iterations with stochastic gradient descent with Nesterov momentum of 0.9, starting with learning rate 0.1 and dropping by a factor of 10 at 32k and 48k steps. We multiply the cross-entropy term by 3 and use weight decay of 0.0001. Compared to the implementation in tensor2tensor library, we include gradient clipping of 1.0 and to create a deep network of 56 layers, we set the number of blocks per layer to 9 for the three blocks. We also set the kernel size to 3 instead of 7 to resemble the original ResNet architecture for CIFAR-10 dataset. For visualization of representations of CIFAR-100 a total of 300 examples were used (100 per class). For the calibration experiment, all 10k examples on the validation set were used to calculate the ECE and reliability diagram. The label smoothing values used when training the teacher in the distillation section were 0.0 to 0.75 with steps of 0.15. For the mutual information results we pick the classes "airplane" and "dog", and calculate the mutual information using 300 examples from the training set of each class. To calculate the average logit per example, we use 5 Monte Carlo samples and estimated mutual information is calculated every 4k training steps. For data augmentation we used random crops and left-right flips as in [6](cifar_image_augmentation).

### A.3   Transformer

The Transformer architecture was used in the calibration section. We reuse the public available implementation by the original authors described in [7]. The dataset can be found in [8](TranslateEndeWmt32k for around 32k tokens). We used the hyper-parameters provided by the authors of original paper for single GPU training (transformer_base for label smoothing of 0.1, and transformer_ls0 for label smoothing of 0.0). Compared to the original paper, we train for 300k steps instead of 100k and we do not use weight averaging of checkpoints. For the calibration experiment, around 10k tokens of the

validation set were used to calculate the ECE and reliability diagram. For calculation of BLEU score (uncased), we use the full validation set [9].

### A.4 Inception-v4

The Inception-v4 architecture was used in the visualization and calibration sections. We reuse a public implementation of the model which be can found at [10]. We modified this code to use a scale parameter for the batch normalization layer, as we found it improved performance. We used a batch size of 4096 and trained the model using stochastic gradient descent with Nesterov momentum of 0.9 and weight decay of $8 \times 10^{-5}$. We took an exponential average of the weights with decay factor 0.9999, and selected the checkpoint that achieved the maximum accuracy during a separate training run where approximately 50,000 images from the ImageNet training set were used for validation.

### A.5 Fully-connected

For the distillation results on MNIST we use fully connected networks with 2 layers. 1200 neurons per layer is used on the teacher and 800 layers in the student. For training the teacher, we use $\alpha = 0.1$, random image shifts of plus or minus 2 pixels in both x and y axis (i.e. 25 equiprobable centerings for each case). The initialization distribution is Gaussian with variance 0.03. Learning rate is set 1, except for last layer which is set to 0.1. We used gradient smoothing corresponding to 0.9 times previous gradient plus 0.1 the current one. Finally the learning rate drops linearly to 0 after 100 epochs and no weight decay is used. For training the student during distillation, no data augmentation is used. We used $\beta = 0.6$, so the original cross-entropy with hard-targets is multiplied by 0.4 and we match the teacher logits with half the squared loss multiplied by 0.6. Note that optimizing for the squared loss is equivalent to picking a high temperature.

## B   Penultimate layer representation for translation

Below, we show visualizations for the English-to-German translation task. The results are similar to the previous image classification visualization results. Next-token prediction is equivalent to classification. In image classification we maximize the likelihood $p(y|x)$ of the correct class given an image, whereas in translation, we maximize the likelihood $p(y_t|x, y_{0:t-1})$ of next token given source sentence and preceding target sequence. However, there are differences between the tasks that affect visualization and distillation.

- The image classification datasets we examine have balanced class distributions, whereas token distributions in translation are highly imbalanced. (This could potentially be addressed with unigram label smoothing.)

- For image classification, we can get near-perfect training set accuracy and still generalize (interpolation regime); in this case, label smoothing erases information, whereas hard-targets preserves it. In the translation task, next-token accuracy on the training set is around 80%. Therefore, visualizations show errors in both the training and validation sets (tending to tight clusters as expected). For distillation, this means a student may learn from the teacher's errors. Since teachers trained with and without label smoothing will both have errors that the student can learn from, it is unclear which will perform best.

- For image classification, the penultimate layer dimension is usually higher than the number of classes, so templates can lie on a regular simplex (i.e. equidistant to each other). In translation, we have a ~30k token alphabet in 512 dimensions, so a simplex is not possible.

In this work, for visualization and distillation, we concentrate on the image classification case to simplify intuition and experimental design. The visualization results below demonstrate that some features of the image classification visualization also hold for translation. We leave analysis of distillation for translation for future work.

Figure 7: Visualizations of penultimate representations of Transformer trained to perform English-to-German translation.

## Footnotes

[2] https://www.tensorflow.org/tutorials/images/deep_cnn

[3] https://github.com/tensorflow/models/blob/master/tutorials/image/cifar10/cifar10.py

[4] https://github.com/tensorflow/tensor2tensor/blob/master/tensor2tensor/data_generators/cifar.py

[5] https://github.com/tensorflow/tensor2tensor/blob/master/tensor2tensor/models/resnet.py

[6] https://github.com/tensorflow/tensor2tensor/blob/master/tensor2tensor/data_generators/image_utils.py

[7] https://github.com/tensorflow/tensor2tensor/blob/master/tensor2tensor/models/transformer.py

[8] https://github.com/tensorflow/tensor2tensor/blob/master/tensor2tensor/data_generators/translate_ende.py

[9] https://github.com/tensorflow/tensor2tensor/blob/master/tensor2tensor/bin/t2t_bleu.py

[10] https://github.com/tensorflow/models/blob/master/research/slim/nets/inception_v4.py