[Reviews · NeurIPS 2019]

Reviewer 1



Label smoothing has become somewhat pervasive in various ML application areas (e.g., MT), but has mostly been thought of as a regularizer. The present work suggests a number of other properties of label smoothing that challenge this assumption, e.g., that label smoothing causes representations to cluster around class templates, that label smoothing improves model calibration, and that label-smoothed networks make worse "teachers" for knowledge distillation. These are important empirical/theoretical findings that are not obvious and should be interesting to a fairly broad audience. The paper is clearly written and the experiments are well designed to support the above claims.

Reviewer 2



Some questions and comments: -- In Figure 4, why does changing the temperature of a network that is already trained with label smoothing degrade calibration? Can the authors offer some insight? -- In Figure 5, why does label smoothing slightly degrade the baseline performance of the student network? Doesn't one expect the student's baseline accuracy to improve by enabling label smoothing? -- From the visualizations shown in the fourth row of Figure 1, it appears like label smoothing could be particularly useful for generalization on samples from classes that are semantically similar. Does this actually hold? (One can examine the confusion matrix of a classification task to see whether confusion between semantically similar classes is resolved in more cases when label smoothing is applied to the model.) -- The visualization idea is neat and it reveals how label smoothing forces training examples of the same class into tight clusters and encourages examples from a class to be equidistant from other classes. This uniformly holds for different datasets and model architectures (rows 1-3 in Fig. 1). While interesting, if we were to rank the contributions of this work, I'd rank this last. I would suggest reorganizing the layout of the paper so that this section on visualization appears after the sections on calibration and distillation. ------- Post rebuttal: Thanks to the authors for addressing my questions. I think this is a strong submission and would really like to see it get accepted. I'm raising my score to an 8.

Reviewer 3



Pros: This paper provides an empirical study to show that label smoothing helps the representation in each class to be close while equally distant to incorrect classes, which is intuitive but can provide insights. Also the effect on reducing the confidence of the prediction and thus help calibration is intuitive and interesting. The author also explains that label smoothing hurts distillation, due to label smoothing erases the relative confidence information between classes and examples, which also make sense to me. Cons: 1. Some of the findings in the paper are somewhat intuitive and natural, such as label smooth reduces confidence and help calibration. 2. What we care more is how these findings help us to get more insights or help us to better use label smoothing or design better methods. 3. The experiments in Section 2 and 4 are only conducted in image classification tasks. I wonder if the phenomenon holds in other tasks in NLP, such as text classification, machine translation. Since the work in mainly based on the empirical study but no theoretical proof, analyses on more tasks are necessary.

[Author Response · NeurIPS 2019]

**Authors' response for paper "When Does Label Smoothing Help?"**

**R1, R2, R3:** Thanks for taking the time to review our paper and for providing thoughtful feedback.

**R2:** *"In Figure 4, why does changing the temperature of a network...with label smoothing degrade calibration?"*:
Temperature scaling is applied after training to correct the calibration of the predictions by dividing the logits by a scalar (temperature), but label smoothing yields calibrated predictions without this additional step. Temperature scaling of an already calibrated network will only degrade ECE.

**R2:** *"In Figure 5, why does label smoothing slightly degrade the baseline performance of the student network?"*: It is hard to separate the effect of one hyperparameter from the many others that affect performance (learning rate schedule, weight decay, etc.), as they are not completely independent. In this experiment, the goal was to compare the relative performance of training the student from label smoothing targets compared to targets provided by a teacher. For fair relative performance comparison, both of these cases have the same training schedule, hyperparameters and equivalent probability mass distributed among incorrect classes. Although in this particular experiment label smoothing hurts performance, in the best performing models in a variety of tasks (Table 1), label smoothing gives consistent gains.

**R2:** *"label smoothing could be particularly useful for generalization on samples from classes that are semantically similar"*: We investigated (same setup as submission) the confusion matrix for the CIFAR-100 dataset, which comprises 20 super-classes containing 5 classes each. Over 5 different runs, we compare the errors made by a network trained with hard targets ($2786 \pm 29$) which can be divided into "fine" errors in the same super-class ($1209 \pm 13$, 43.4% of total errors) and "coarse" errors in a different super-class ($1577 \pm 30$). With label smoothing, the total # of errors is reduced ($2732 \pm 34$) with a reduction of "coarse" errors ($1515 \pm 25$) while the "fine" errors remains close to constant ($1217 \pm 17$) representing 44.6% of the total errors. While reduction of only coarse errors is not intuitive, we observe that the ratio of "fine" errors remains stable and we believe more experiments should be done to verify if the effect is consistent.

**R2:** *"I would suggest reorganizing the layout of the paper"*: We agree that the effect of label smoothing on calibration and distillation has direct practical applications, but we also feel the visualization provides useful intuition. That said, we are open to revisiting the ordering in the camera-ready version.

**R3:** *"Conduct more empirical analysis of other tasks to verify the findings on label smoothing"*: Below, we show visualizations for the English-to-German translation task. The results are similar to the image classification visualization results in the submission. Next-token prediction is equivalent to classification: In image classification we maximize the likelihood $p(y|x)$ of the correct class given an image, whereas in translation, we maximize the likelihood $p(y_t|x, y_{0:t-1})$ of next token given source sentence and preceding target sequence. However, there are differences between the tasks that affect visualization and distillation.

- The image classification datasets we examine have balanced class distributions, whereas token distributions in translation are highly imbalanced. (This could potentially be addressed with unigram label smoothing.)

- For image classification, we can get near-perfect training set accuracy and still generalize; in this case, label smoothing erases information, whereas hard-targets preserves it. In the translation task, next-token accuracy on the training set is around 80%. Therefore, visualizations show errors in both the training and validation sets (tending to tight clusters as expected). For distillation, this means a student may learn from the teacher's errors. Since teachers trained with and without label smoothing will both have errors that the student can learn from, it is unclear which will perform best.

- For image classification, the penultimate layer dimension is usually higher than the number of classes, so templates can lie on a regular simplex (i.e. equidistant to each other). In translation, we have a ~30k token alphabet in 512 dimensions, so a simplex is not possible.

In our submission, for visualization and distillation, we concentrate on the image classification case to simplify intuition and experimental design. The visualization results below demonstrate that some features of the image classification visualization also hold for translation. We leave analysis of distillation for translation for future work. We will include a section dedicated to the differences between these cases in the appendix, as well as the figure we provide here.

Figure 1: Visualizations of penultimate representations of Transformer trained to perform English-to-German translation.

[Meta-Review · NeurIPS 2019]

In this paper, the authors provide a comprehensive empirical study on label smoothing for deep learning. The study contains quite a few interesting insights, e.g., label smoothing implicitly calibrates learned models by making their predictions more consistent with the underlying accuracy; while teacher networks trained with label smoothing get more accurate, they are ultimately less effective at distilling knowledge into a student network. The reviewers raised some concerns, including the relatively narrow scope of the experiments, and the lack of theoretical analysis. The authors did a good job in rebuttal, and all the reviewers agree after reading the rebuttal that the empirical insights in this paper have high practical value, and NeurIPS audience should benefit from them